# Alkali-Induced Hydrolysis Facilitates the Encapsulation of Curcumin by Fish (*Cyprinus carpio* L.) Scale Gelatin

**DOI:** 10.3390/foods14071183

**Published:** 2025-03-28

**Authors:** Jia Liu, Wan Aida Wan Mustapha, Xiaoping Zhang, Haoxin Li

**Affiliations:** 1Guizhou Academy of Agricultural Sciences, Guiyang 550006, China; mcgrady456@163.com; 2Key Laboratory of Environmental Pollution Monitoring and Disease Control, School of Public Health, Ministry of Education, Guizhou Medical University, Guiyang 561113, China; 3Department of Food Sciences, Faculty of Science and Technology, Universiti Kebangsaan Malaysia (UKM), Kuala Lumpur 43600, Malaysia; wanaidawm@ukm.edu.my; 4Guizhou Fishery Research Institute, Guizhou Academy of Agricultural Science, Guiyang 550025, China; tesslhx1212@gmail.com

**Keywords:** alkali hydrolysis, fish scale gelatin, hydrophobic interaction

## Abstract

Curcumin-loaded alkali-induced fish scale gelatin (AFSG) was fabricated to evaluate its efficacy as a potential carrier for hydrophobic nutrients. In this study, the effect of the alkali hydrolysis period on the AFSG hydrolysate structure and corresponding curcumin loading efficiency have been elucidated. Results showed that alkali-induced degradation of gelatin yields different polymers with molecular weights (*M_w_*) from 19319 to 3881 Da. Moderate alkali hydrolysis of fish scale gelatin exposes hydrophobic amino acids, enhancing hydrophobic interactions and increasing the proportion of these amino acids. This process also promotes a structural shift, favoring β-sheet formation while reducing α-helix content. Moreover, the curcumin loading efficiency of AFSG (2 h) (10.06 ± 0.27 μg/mL) was significantly higher than that of untreated gelatin (2.16 ± 0.39 μg/mL), while its excessive hydrolysis weakens hydrophobic interactions among hydrophobic amino acids, limiting their binding sites for curcumin. Fluorescence spectroscopy indicated that curcumin-induced fluorescence quenching in AFSG follows a static mechanism. Thus, the above results demonstrated AFSG’s potential as an effective carrier for lipophilic nutrients with high encapsulation efficiency.

## 1. Introduction

Curcumin, a lipophilic polyphenolic compound derived from Curcuma longa rhizomes, is a rare diketone structure exhibiting diverse pharmacological properties and beneficial health effects [1], such as anti-cancer [2], anti-inflammatory [3], and antioxidant activities [4], etc. However, its low water solubility, limited bioavailability, rapid intestinal metabolism, and susceptibility to degradation and oxidation during processing and storage limit its broader application in functional foods and pharmaceutical formulations [1,5,6]. In addition, the challenges mainly stem from delivering this bioactive substance into a suitable carrier, making their integration into food systems a significant technical hurdle in the food industry.

Nanocarrier delivery systems hold significant promise for addressing these challenges by enhancing the bioavailability, stability, and solubility of curcumin in food applications. The embedding loading technology based on biomacromolecules can effectively realize the steady state of these hydrophobic active substances [7]. Natural biopolymer-based nanocarriers, such as proteins [8], liposomes [9], and carbohydrates [10,11], are widely employed for curcumin delivery. Among them, proteins extracted from a variety of natural sources can be used as promising nano-delivery carriers in foods because of their ability to possess a natural hydrophobic region and serve as a carrier for small hydrophobic molecules without requiring chemical modification [8]. Gelatin was chosen for its natural, biocompatible properties, derived from heat-denatured collagen through acidic or alkaline processes. It is extensively applied in the food and pharmaceutical sectors due to its multifunctional properties [12]. Notably, mammalian sources contribute to approximately 98.5% of global gelatin production [13]. However, the safety of mammalian gelatin has been increasingly questioned due to its potential role in transmitting prion-related pathogens, particularly in the wake of diseases such as bovine spongiform encephalopathy (BSE) and foot-and-mouth disease (FMD) [14]. In response to these concerns, global consumers with strict adherence to vegetarianism and religious beliefs seek alternative protein sources with similar functionalities for food systems [12]. As a result, there has been growing interest in alternative gelatin sources, particularly those derived from freshwater fish processing by-products such as scales, bones, and skin. Compared to mammalian gelatin, these alternatives not only eliminate concerns associated with prion-related diseases but also better align with the dietary preferences and ethical considerations of consumers adhering to vegetarianism and religious restrictions. Although freshwater fish may be susceptible to certain microbial contaminants, appropriate processing and purification methods can effectively mitigate these risks, ensuring its safety and suitability as a viable alternative to mammalian gelatin in food applications [12].

However, fish gelatin exhibits weaker gelling and rheological properties than mammalian gelatin, largely due to its reduced proline content, hydroxyproline, and amino acids, limiting its large-scale application [7]. To this end, this has driven the necessity to overcome these limitations, prompting the development of modification methods to enhance its functional properties, such as genipin and glutaraldehyde [15,16], MTGase [17], and k-carrageenan [18,19]. However, considering that these chemical crosslinking agents or organic solvents are still involved in the encapsulation process of bioactive substances, alkali hydrolysis is a straightforward and easily adaptable method used to modify protein structure and, consequently, its functional properties. This approach is favored for its appealing characteristics, such as safety, low energy requirements, and the absence of organic solvents in self-assembly. During alkali hydrolysis, the protein solution is maintained at an alkaline pH for a specific duration. Shifting the pH from the isoelectric point increases charge repulsion between protein molecules. This alteration disrupts the native conformation of the protein and subsequently impacts its functional properties [19]. While curcumin is widely used in culinary applications, its inherent limitations—such as rapid photodegradation, poor aqueous solubility (~11 ng/mL in water), and thermal instability during cooking processes—severely restrict its bioavailability and industrial applicability [20]. Thus, we hypothesized that the encapsulation within alkali-induced fish scale gelatin (AFSG) offers a promising strategy to enhance curcumin’s dispersibility in aqueous environments while providing structural protection against degradation. By investigating the encapsulation of curcumin within AFSG, this study provides insights into an alternative delivery system that extends curcumin’s applications beyond conventional culinary use.

Although self-assembly induced by alkali-induced fish scale gelatin (FSG) is of both fundamental and practical importance, there is still limited information on its applications. The specific role of protein structure in the alkali-induced hydrolysis of fish scale gelatin (AFSG) remains inadequately explored. Therefore, this study investigates the effects of varying alkali hydrolysis durations upon the structural and functional properties of AFSG, along with its interaction with curcumin. Fourier transform infrared spectroscopy (FTIR), X-ray diffraction (XRD), fluorescence spectra, and dynamic light scattering (DLS) were employed to characterize the self-assembly behavior of AFSG with and without curcumin loading. Additionally, confocal laser scanning microscopy (CLSM) and transmission electron microscopy (TEM) were utilized to investigate the morphological structure, photochemical properties, and interactions between curcumin and AFSG. This study investigates the self-assembly behavior of alkali-treated fish scale gelatin and its suitability as a delivery system for hydrophobic bioactive compounds, offering a scalable approach to enhance FSG utilization in food applications.

## 2. Materials and Methods

### 2.1. Chemicals and Materials

Fresh carp (*Cyprinus carpio* L.) scales were collected from the experimental base of the Guizhou Fisheries Research Institute in Guiyang, Guizhou Province, China. They were washed three times with tap water to remove impurities, then dried in an oven (model WGL-65B, Taisite Instrument Co., Ltd., Tianjin, China) at 40 °C for 24 h to remove excess water. The resultant product was stored at 15 ± 0.5 °C in a desiccator until analysis. Sodium chloride (NaCl), sodium hydroxide (NaOH), hydrochloric acid (HCl), potassium bromide (KBr) of infrared (IR) spectroscopy grade, and β-mercaptoethanol (β-ME) were obtained from Macklin Biochemical Co., Ltd. (Shanghai, China). Coomassie Brilliant Blue was bought from Servicebio Biotechnology Co., Ltd. (Wuhan, China). Curcumin (98%) was supplied by Adamas Reagent Co., Ltd. (Shanghai, China) and 8-anilino-1-napthalenesulfonic acid ammonium salt (ANS) was acquired from Wanbang Chemical Technology Co., Ltd. (Nanjing, China).

### 2.2. Preparation of Alkali-Induced Fish Scale Gelatin (AFSG) and Curcumin-Loaded Nanoparticles (CL-AFSG)

Fish scale gelatin was extracted following the method outlined in our previous study [7]. Alkali-induced fish scale gelatin (AFSG) was prepared by treating samples with sodium hydroxide (0.1 M) for 0 to 8 h, followed by neutralization to pH 7 using HCl (0.1 M). The resultant solution was dialyzed (MWCO: 14 kDa) overnight to eliminate surplus alkali ions, followed by lyophilization. The processed samples were subsequently refrigerated at 4 °C for further analysis.

To prepare curcumin-loaded nanoparticles, AFSG (40 mg) was initially dissolved in 20 mL of distilled water by boiling for 1 min [21]. Following full dissolution, the solution was cooled to 15 ± 1 °C and adjusted to 20 mL with deionized water. Subsequently, 1 g of curcumin powder was blended into the AFSG solution and homogenized (12,000 rpm, 2 min) using an IKA T18 basic homogenizer (IKA-Werke GmbH & Co., Staufen, Germany). To ensure adequate interaction between curcumin and AFSG, the mixture was homogenized and stirred at 300 rpm in a water bath maintained at 30 ± 1 °C for 24 h. The sample was subsequently centrifuged at 5600× *g* for 10 min, and the supernatant was gathered for subsequent analysis. The determination of curcumin concentration of CL-AFSG solution was measured using HPLC as in our previous method [21]. The curcumin loading efficiency (CLE, µg/mL) was calculated as the amount of curcumin (µg) present in 1 mL of AFSG solution, determined using an external calibration curve.

### 2.3. Determination of M_w_ Among AFSGs

The sample powder of FSG, AFSG_2h_ (duration of acid hydrolysis = 2 h), and AFSG_8h_ (duration of acid hydrolysis = 8 h) were acquired by vacuum-freeze drying (10YG/A, SCIENTZ Co., Ltd., Ningbo, China) at −60 °C for 12 h and used for subsequent characteristic analysis.

The determination of *M_w_* among FSG and AFSG was performed with a high-performance size exclusion chromatography (HPSEC) system, which consisted of a gel column (TSKgel 7.8 × 300 mm G5000PWXL, Tosoh Bioscience GmbH, Griesheim, Germany) and an RI detector (10A, Shimadzu, Kyoto, Japan) [22]. The calculation of the apparent *M_w_* for the samples was based on calibration with glucan standard reference (*M_w_* of 4 × 10^3^, 7 × 10^3^, 50 × 10^4^, 100 × 10^4^, and 200 × 10^4^ g/mol). The calibration curve of these standard samples is given in the Appendix A (Appendix A).

### 2.4. Amino Acids Analysis of AFSGs

10 mL of 6 M HCl was added to a glass hydrolysis tube containing 200 mg of AFSGs. The hydrolysis was performed by using a drying oven at 110 ± 2 °C for 24 h. After hydrolysis, it was cooled to room temperature and filtered into a 50 mL volumetric flask with a 0.45 μm nylon membrane. After absorbing 2 mL of the sample, the mixture solution was dried under vacuum and fully dissolved with 1 mL of 0.02 M HCl on the rotary evaporator for deacidification at 45 °C. Then, 2 mL of sample buffer was added to fully dissolve and was then analyzed by an amino acid analyzer (A300, membraPure GmbH, Hennigsdorf, Germany).

### 2.5. Circular Dichroism Spectroscopy (CD) Analysis

The secondary structures of the samples (0.5 mg/mL) were analyzed using a CD spectropolarimeter (JASCO J-815, JASCO Co., Ltd., Tokyo, Japan). Scanning was performed in the far-infrared wavelength range from 190 nm to 250 nm at 1 nm intervals.

### 2.6. Surface Hydrophobicity (H_0_)

The H_0_ of samples was determined using an 8-anilino-1-napthalenesulfonic acid ammonium salt (ANS) as a fluorescent probe. Each sample dispersion was diluted with distilled water to obtain a sequence of protein concentrations (0.0025%, 0.005%, 0.01%, 0.02% *w/w*), and was added into acrylic cuvettes (Sarstedt Inc., Nümbrecht, Germany) together with 40 μL of ANS reagent (8 mM). The fluorescence intensity (FI) of these samples was measured by a luminescence spectrometer LS50B (PerkinElmer, Waltham, MA, USA) at an excitation wavelength (λ_Ex_) of 390 nm and an emission wavelength (λ_Em_) of 470 nm. The slope of the entirely linear curve of FI vs. protein concentration was used as the indicator of H_0_.

### 2.7. Measurement of the Intermolecular Interactions

The samples were treated with denatured solvents that could destroy different interactions between molecules, following the method described before [23]. Sodium chloride solution (0.6 mol/L) could disrupt ionic bonds, 1.5 mol/L urea could destroy hydrogen bonds, 8 mol/L urea could simultaneously destroy the hydrogen bonds and hydrophobic interactions, and 0.5 mol/L β-mercaptoethanol (β-ME) could disrupt disulfide bonds. Samples were solubilized in four selected solvents, respectively, to analyze the contribution of different interactions in the formation process of the fermentation-induced gel. The four selected solvents were: 0.6 mol/L sodium chloride solution (Sol1), 0.6 mol/L sodium chloride solution + 1.5 mol/L urea (Sol2), 0.6 mol/L sodium chloride solution + 8 mol/L urea (Sol3), and 0.6 mol/L sodium chloride solution + 8 mol/L urea +0.5 mol/L β-mercaptoethanol (Sol4). The sample (1 g) was dissolved in 10 mL of the different solvents, homogenized, and then centrifuged (4000× *g*, 4 °C, 15 min). Using BSA as the standard, the protein concentration in the supernatant was determined by the Coomassie brilliant blue method. The solubility of Sol 1, (Sol2-Sol1), (Sol3-Sol2), and (Sol4-Sol3) represent the contribution of ionic bonds, hydrogen bonds, hydrophobic interactions, and disulfide bonds, respectively, and is expressed as soluble protein/homogenate (mg/mL).

### 2.8. AFM Determination

The nanostructure characterization of alkali-treated gelatin samples was further performed using a Multimode 8 AFM (Dimension Icon, Bruker, Germany) in tapping mode. Briefly, 10 μL of sample aqueous suspension (2.0 mg/mL) was added onto a freshly cleaved mica (EMS, Hatfield, PA, USA) surface. All samples were measured after drying at ambient temperature. All the images were scanned in the air using standard peak force-mode silicon RTESP-150 cantilevers (parameters of the AFM cantilever were T: 1.75 μm; L: 125 μm; W: 35 μm; f_0_: 150 kHz; k: 6 N/m). Images were obtained using both the height mode and the error signal mode. The height mode includes both 3-dimensional and 2-dimensional images. The error signal mode removed slow variations in the surface topography but highlighted the edges of the features [24]. To optimally capture distinct morphological features of the samples, the scanning rate in auto-mode was dynamically adjusted based on scan size, ensuring a constant pixel resolution of 512 × 512 across all images [25]. The sample dimensions (height) of the observed aggregates were analyzed by Nanoscope Analysis software (version 1.8).

### 2.9. Characterization of AFSG with or Without the Loading of Curcumin

#### 2.9.1. Fourier Transform Infrared Spectroscopy (FT-IR)

FTIR spectra were employed to assess the molecular interaction forces among freeze-dried AFSGs with or without the loading of curcumin and their physical mixture. The lyophilized sample (1 mg) and potassium bromide powder (100 mg) were mixed, ground, and tableted before spectra acquisition. Fourier transform infrared spectra were recorded by the FTIR 2000 spectrophotometer (Perkin-Elmer, Norwalk, CT, USA). The resolution of the spectrometer was 4 cm^−1^, the number of iterations was 32, the scanning speed was 5 kHz, the sensitivity was 1, and the spectra were collected in a spectral range of 500–4000 cm^−1^ [17].

#### 2.9.2. X-Ray Diffraction

XRD patterns (from 6 to 30°) of freeze-dried powders of curcumin, FSG and AFSG, curcumin to FSG and AFSG (1:50, *w/w*) physical mixture, and curcumin-loaded FSG and AFSG were analyzed with an X-ray polycrystalline powder diffractometer (D8 ADVANCE, Brugger, Germany) at a scanning speed of 2 °/min with the voltage of 40 KV.

#### 2.9.3. TEM Observation

The morphology of AFSGs with and without the loading of curcumin was characterized by transmission electron microscopy (TEM) (JEM 1200EX, JEOL Ltd., Akishima, Tokyo, Japan) [21]. The freshly prepared sample was diluted 10 times with deionized water. Then, one drop of a diluted sample was placed on a freshly glow-discharged carbon film on a 400-mesh copper grid and stained with 1% phosphate tungsten. The observation was imaged with a Tungsten filament lamp at 100 kV and 15,000× magnification.

#### 2.9.4. Particle Size, ζ-Potential, and Polydispersity Index (PDI) Measurements

A Zetasizer Nano ZS90 (Samufei instruments, Houston, TX, USA) equipped with a He-Ne laser (633 nm) and a scattering angle of 165° was employed for the determination of particle size, polydispersity index (PDI), and ζ-potential of samples [21]. The intensity-based mean size, ζ-potential, and polydispersity index (PDI) were determined based on dynamic light scattering (DLS). The ζ-potential was determined on the ground of the particle surface charge. The temperature during the particle size and ζ-potential measurement was 25 ± 0.5 °C and expressed as nm and mV. Each experiment was repeated three times.

### 2.10. Fluorescence Spectroscopy

The fluorescence spectra were exploited to characterize the interaction between FSG, AFSG (alkali hydrolysis for 2 h and 8 h), and curcumin using an F7000 fluorescence spectrophotometer (Hitachi High-Tech Co., Tokyo, Japan) according to the previous method [26]. The measurement was performed at an excitation wavelength of 280 nm, and the resulting emission spectra were collected at a wavelength ranging from 290 nm to 370 nm. The emission light slit is 5 nm, and the excitation light slit is 10 nm. All the measurements were performed in triplicate at room temperature. The intrinsic fluorescence emission quenching of FSG or AFSG with the addition of curcumin (0, 1, 2, 3, 4, 5, and 6 μmol/L) was analyzed using the Stern-Volmer equation as follows [27]:(1)$$\frac{F_{0}}{F}=1+K_{sv\;}[Q]=1+K_{q\;}\tau_{0\;}[Q]$$

Static quenching can be described by providing information on the binding constant K_a_ and the number of binding sites using the following formula [28]:(2)$$Log\left(\frac{F_{0}}{F}-1\right)=LogK_{a}+nLog[Q]$$
where F_0_ is the fluorescence intensity of AFSG without quencher; F is the fluorescence intensity of FSG with quencher; [Q] is the concentration of the quencher; k_q_ is the quenching rate constant; τ_0_ is the average fluorescence lifetime of biomacromolecules without quencher and equals to 10^−8^ s; K_a_ is the binding constant; n is the number of binding sites.

### 2.11. Confocal Laser Scanning Microscopy (CLSM)

The spatial distribution of curcumin-loaded nanoparticles in native and alkali-modified FSG matrices was analyzed via CLSM (LEICA TCS SP8) following established protocols [25]. Fluorescence detection parameters were configured as follows: Curcumin fluorescence was detected at 470–556 nm (green channel) under 405 nm excitation; rhodamine B-labeled gelatin samples (both native and alkali-treated FSG) were analyzed using 570 nm excitation with emission collected at 565–685 nm (red channel). Following standardized preparation, samples were deposited on 1 mm glass substrates and equilibrated at 10 °C for 12 h prior to imaging. Image acquisition was conducted using a HyD high-sensitivity detector equipped with argon-ion and He-Ne laser.

### 2.12. Statistical Analysis

The data were expressed as mean ± SD values. All data were analyzed by the statistical software statistical package for the Social Sciences 20.0 (SPSS, Chicago, IL, USA). The one-way analysis of variance (ANOVA) was performed to determine the least significant at *p* < 0.05 by Tukey’s HSD test. All graphics were plotted using Origin software 2018 (OriginLab Corporation, Northampton, MA, USA).

## 3. Results

### 3.1. Effect of Molecular Weight (M_w_) and Degree of Alkali Hydrolysis on the CLE (Curcumin Loading Efficiency, µg/mL) of AFSG

The impact of molecular weight on designing nanoparticle delivery systems for curcumin encapsulation was examined by measuring the curcumin loading efficiency (CLE) of AFSG through HPLC. As depicted in Figure 1A, native FSG encapsulated a minimal amount of curcumin (2.16 ± 0.39 μg/mL) in its neutral hydrophobic region, likely due to the presence of hydrophobic amino acids such as proline, alanine, and phenylalanine within native FSG (Table 1). Fish scale gelatin’s limited CLE may result from the higher abundance of hydrophilic groups (-COOH and -NH^3+^) resulting from the breakdown of peptide bonds. Noticeably, as shown in Figure 1B, alkali hydrolysis in the initial stage resulted in a notable decrease in the *M_w_* of native fish scale gelatin, decreasing from 60,906 g/mol (0 h) to 19,319 g/mol (1 h) and 16,450 g/mol (2 h).

The results in Table 1 could be explained by the fact that the tendency of gelatin is to undergo degradation in highly alkaline conditions [29]. Meanwhile, on the other hand, in terms of the CLE of AFSG, at the initial stage, its value was markedly greater (*p* < 0.05) than that of the native gelatin, increasing from 6.11 ± 0.45 (1 h) to 10.06 ± 0.27 μg/mL (2 h). Compared to the aqueous solubility of curcumin powder (11 ng/mL) [8], curcumin solubility in AFSG_2h_ exhibited a dramatic improvement, increasing by approximately 1105-fold. This could be attributed to the fact that the alkali hydrolysis of FSG would give rise to the degradation of gelatin with decreased *M_w_*. This process exposed more hydrophobic regions as the molecular skeleton broke down, facilitating the formation of robust hydrophobic cores within the core-shell structure. Consequently, this enhanced structure provided an ideal environment for encapsulating a greater amount of curcumin. Nevertheless, with extended alkali hydrolysis of FSG, a gradual decline in curcumin loading efficiency (CLE) was observed, ranging from 4.73 ± 0.18 to 8.36 ± 0.24 μg/mL (Figure 1A). This trend suggests that the continued decrease in *M_w_* enhanced molecular mobility, thereby disrupting the formation of stable intramolecular hydrophobic regions [14,30]. Furthermore, the reduction could also be linked to the hydrophobic aggregation of FSG or the degradation of curcumin binding sites due to prolonged alkali hydrolysis, leading to diminished curcumin binding capacity.

### 3.2. Effect of Alkali Hydrolysis on Amino Acid Content of AFSG

The amino acid composition of AFSGs was analyzed to determine the ratio of hydrophobic amino acids to the total (Table 1). Tryptophan was undetectable due to its degradation during acid hydrolysis, while glutamine and asparagine were transformed into aspartic acid and glutamate as a result of the hydrolysis process. A notable observation was the significant reduction in total amino acid (TAA) content in AFSG compared to FSG. Specifically, the TAA content of AFSG_2h_ and AFSG_8h_ was 489.02 and 483.55 mg/g, respectively, representing a decrease of 6.7% and 7.8% from FSG’s 524.21 mg/g. This decline can be attributed to alkali hydrolysis, which caused peptide bond cleavage and amino acid degradation, leading to the release of some proteins into the solution. Regarding the HAA-to-AA ratio, it showed a slight increase from 41.13% (0 h) to 42.13% (2 h) and 41.45% (8 h), suggesting that alkali treatment promotes FSG unfolding and conformational relaxation. This process generates more peptides with C-terminal hydrophobic amino acid residues compared to untreated FSG hydrolysis, thereby enhancing hydrophobic amino acid exposure. To this end, it can be inferred that alkali treatment exposes hidden hydrophobic regions, facilitating the formation of self-assembled nanoparticles that serve as an effective carrier for curcumin.

### 3.3. Effect of Alkali Hydrolysis on Surface Hydrophobicity and Secondary Structure Composition of AFSG

The surface hydrophobicity of proteins is widely recognized as a crucial factor influencing their conformational structure and functional properties [31]. As shown in Figure 2A, regarding natural FSG, hydrophobic groups embedded within the core of the folded structure contribute to the low surface hydrophobicity, indicating that FSG has limited hydrophobic binding sites available for ANS binding. However, following alkali hydrolysis, AFSG_2h_ demonstrated the greatest surface hydrophobicity, trailed by AFSG_8h_, suggesting that the hydrolysis process modified the probe’s binding sites within the peptides. The exposure of hydrophobic moieties increased in H_0_. The elevated surface hydrophobicity observed in AFSG_2h_ can be attributed to the FSG partial degradation, which increased the exposure of hydrophobic regions and facilitated the formation of soluble aggregates. In addition, AFSG_2h_ offered additional binding sites for curcumin, enabling greater curcumin binding via hydrophobic interactions, a result that aligns with the observed curcumin loading efficiency (CLE) data. On the contrary, excessive hydrolysis resulted in protein denaturation and increased soluble aggregates.

Circular dichroism (CD) is a highly versatile and extensively employed spectroscopic method for studying protein structures, offering precise insights into molecular-level conformational changes, with a particular focus on secondary structure analysis [32]. To further elucidate the impact of alkali treatment on FSG’s secondary structure, the CD spectra of samples were analyzed within the 190 nm to 250 nm range. As depicted in Figure 2B, the CD spectra of gelatin showed two distinct peaks: a negative peak at 205 nm and a positive peak at 222 nm [33]. The proportions of secondary structural elements, such as α-helix, β-turn, β-sheet, and random coil, were determined from the spectra and are presented in Figure 2C. At the initial stage of alkali hydrolysis, AFSG_2h_ had an increased *β*-sheet with a concomitant decrease in α-helix structure. Generally, the α-helix is stabilized by intramolecular hydrogen bonds formed between the carbonyl oxygen (CO) and the amino hydrogen (NH) of the peptide chain [34]. The decreased α-helix might offer clues about the weakened hydrogen bonding, and the molecular structure progressively shifted from an ordered arrangement to a disordered state, as detailed in Appendix A. Besides, lower α-helix contents in AFSGs (3.2~3.5%) than in native FSGs (12.8%) demonstrated increased surface hydrophobicity and a looser structural arrangement. Furthermore, the *β*-sheet structure, known for its higher flexibility and openness relative to the α-helix [35], suggests that early-stage alkali treatment (2 h) induced a more flexible and adaptable FSG conformation. Furthermore, the observed changes in secondary structure content likely stemmed from the breakdown of internal molecular cohesion and intermolecular interactions caused by denaturation during alkali hydrolysis, which may have disrupted intramolecular hydrogen bonds, ultimately leading to alterations in the secondary structure. With prolonged alkali hydrolysis, the *β*-sheet content progressively decreased, while there was a slight increase in the random coil content, indicating possible reaggregation of the unfolded FSG. Extended hydrolysis (8 h) likely weakened hydrophobic domains and disturbed the intermolecular forces’ balance, destabilizing and eventually disrupting the *β*-sheet structure in protein aggregates.

### 3.4. Effect of Alkali Hydrolysis on Intermolecular Interaction of AFSG

To gain deeper insights into the influence of intermolecular forces on protein structure during alkali hydrolysis, the intermolecular interactions were assessed. The addition of various denaturants to proteins can disrupt intermolecular forces—such as hydrophobic interactions, hydrogen bonds, ionic bonds, and disulfide bonds—that stabilize the gel, thereby enhancing protein solubility. As illustrated in Figure 3, hydrophobic interactions were significantly strengthened, and disulfide bonds were slightly reinforced in AFSG_2h_, suggesting that during the initial phase of alkali hydrolysis (0–2 h), hydrophobic groups and oxidized sulfhydryl groups were partially exposed. At this stage, the protein exhibited increased molecular flexibility and a tendency to aggregate. Compared to native FSG, AFSG_2h_ displayed a more expanded structure, which, supported by hydrophobic interactions, facilitated the formation of a stable configuration, highlighting the role of hydrophobic forces in maintaining the structure of alkali-treated FSG. Moreover, the proportion of sulfur-containing amino acids (methionine and cysteine) in AFSG_2h_ was notably higher compared to other samples; thus, these free sulfhydryl groups were combined to accelerate the formation of disulfide bonds. However, excessive alkaline hydrolysis caused excessive denaturation of protein regarding AFSG_8h_, resulting in excessive cross-linking and aggregation among protein molecules, thus forming an unstable structure. On the other hand, alkali hydrolysis of FSG led to the impaired hydrogen bond and ionic bond, large sections of the *β*-sheet were broken down into smaller, dispersed *β*-sheets pieces, and fewer hydrogen bonding sites were required; thus, the hydrogen bond weakened.

### 3.5. AFM Observation

To investigate whether the secondary structural transformation is accompanied by changes in nanoscale morphology, we examined the effect of extending the alkali hydrolysis period on the particle morphology of FSG and AFSGs using AFM. The height mode, featuring both 2D and 3D topographical representations, along with error-signal mode images, is illustrated in Figure 4A–I. The native FSG nanoparticle had heterogeneous and irregular coacervate structures, including the coexisting rod-like and spheroid. Likewise, gelatin aggregation proceeds through a multimeric association (or cluster association) mechanism, in which multimers of diverse sizes assemble into cluster structures, which then further coalesce into spherical aggregates [36]. With the extension of alkali hydrolysis, AFSG samples tended to form isolated aggregates, with AFSG_2h_ exhibiting a more homogeneous and uniform particle morphology featuring annular pores. Consistently, studies have shown that annular pores form during the alkaline pretreatment of gelatin from channel catfish skins [24], likely due to the uneven penetration of alkaline solutions into gelatin molecules during hydrolysis. Except for the change of morphological characteristics after alkali hydrolysis, the size of the AFSG particles was much smaller. To accurately grasp the particle information, particle analysis was performed using Nanoscope Analysis software 3.00 to generate size histograms of height distribution. Figure 4J revealed that the larger FSG aggregates broke down into smaller particles following alkali hydrolysis, suggesting significant alterations in FSG particle morphology owing to the alkali treatment. In detail, the FSG structure featured a prominent aggregated particle with a height of 266 nm. Prominently, having undergone the alkali hydrolysis at the initial stage, the height of AFSG_2h_ decreased to 36.9 nm. With the continued prolongation of alkali hydrolysis, the height of AFSG_8h_ was further reduced, resulting from the loosened structure of FSG caused by excessive alkali treatment. As the loose subunits continued to dissociate completely, the particle height decreased accordingly.

### 3.6. SEC Profiles

To unravel the aggregation behavior of FSG post-alkali treatment, the SEC profiles of both native and alkali-treated FSG were analyzed (Figure 5). Native FSG displayed distinct elution peaks at 14.62 min, whereas alkali treatment resulted in significant alterations in the elution profiles, characterized by the emergence of exclusion peaks around 18.22 min during the initial hydrolysis phase. This indicated that a portion of the fractions in FSG samples underwent alkali treatment for the initial 2 h, transforming into lower molecular weight soluble aggregates. Moreover, the characteristic elution peak area of the FSG sample located at 14.62 min when treated by alkali hydrolysis gradually decreased and shifted to 15.28 (AFSG_2h_) and 15.74 (AFSG_8h_) min, respectively.

With the further prolongation of alkali hydrolysis, the characteristic elution peak area of the FSG eventually disappeared, suggesting that FSG molecules progressively denatured and engaged in polymerization. Conversely, the fraction of the elution peak area between 16–18 min rose as the FSG-related peak diminished, implying that prolonged alkali hydrolysis, especially toward the end, tended to promote the growth of protein aggregates and the formation of smaller soluble aggregates.

Consequently, as alkali hydrolysis progresses, protein molecules tend to aggregate and gradually form small-molecular-weight soluble aggregates, which aligns with the observed particle size distribution data. A smaller amount of material was eluted within the 15–18-min range after 2 h of alkali treatment compared to other samples, suggesting the aggregates were smaller in size post-treatment. In summary, the SEC results suggested that AFSG_2h_ experienced milder denaturation compared to other AFSG samples, whereas alkali hydrolysis of FSG for 8 h resulted in the denaturation of FSG. Combining the findings from H_0_ and TEM, it was inferred that alkali treatment promotes the reaggregation of particles into smaller soluble aggregates, primarily driven by S–S bonds and hydrophobic interactions.

### 3.7. Self-Assembly Behavior of FSG and AFSG with or Without Loading Curcumin

The self-assembly properties of FSG and AFSG, both with and without curcumin loading, were further investigated through combined analysis, including TEM and DLS. FSG nanoparticles exhibited a particle size of 1266.3 nm, with a PDI of 0.824 (Figure 6A), aligning with TEM observations that showed grainy FSG with a rough surface, large particle size, and a tendency to aggregate. At the initial stage of alkali hydrolysis, the self-aggregated nano-system of AFSG_2h_ was relatively small and more homogenously distributed than native FSG with an evident core-shell structure. However, with further extension of alkali hydrolysis, the emergence of additional hydrophobic sites and the formation of new aggregates drove substantial aggregation, leading to increased particle size and a decreased PDI, as supported by the TEM findings. On the other hand, Figure 6C exhibited that after loading curcumin, all CL-AFSG samples formed uniformly distributed spherical particles, as indicated by the reduced particle size and PDI. Noticeably, regarding morphology, CL-AFSG_2h_ smooth-surfaced spherical particles, with the majority measuring less than 400 nm in diameter, are consistent with the DLS results (Figure 6A). Given the above results, it is well suggested that moderate alkali hydrolysis of FSG enhances the exposure and integration of hydrophobic groups during the reassembly process, promoting the formation of uniform, spherical nanoparticles. This process significantly improves curcumin encapsulation by facilitating stronger hydrophobic interactions. However, after excessive alkali hydrolysis of FSG (AFSG_8h_), TEM images displayed distinct block-like aggregates, where the overexposure of the protein’s hydrophobic moieties prompted hydrophobic aggregation mediated by the protein itself.

Meanwhile, coupled with the decreased particle size, the inward curling of negatively charged groups reduced the absolute ζ-potential value. The zeta potentials of FSG, AFSG_2h_, and AFSG_8h_ were −25.2, −14.5, and −12.3 mV, respectively (Figure 6B). Consequently, the weaker electrostatic repulsion between AFSG_8h_ particles indicates a stronger inclination for aggregation, which accounts for the larger particle size observed in AFSG8h [8]. However, the incorporation of curcumin into native and alkali-treated FSG led to a decrease in both particle size and PDI, as the reinforced hydrophobic interactions facilitated the formation of more compact and uniformly structured particles. Notably, the PDI of CL-AFSG_2h_ was approximately 0.2, indicating a highly uniform size distribution and excellent dispersibility. These findings suggest that controlled alkali hydrolysis enhances FSG self-assembly, while fortified hydrophobic interactions further promote the formation of smaller, more stable nanoparticles.

### 3.8. Interaction Between Curcumin and AFSG

#### 3.8.1. FTIR Analysis

To explore the mechanism underlying the improvement in curcumin loading efficiency from alkali-induced FSG, FTIR spectroscopy was employed to monitor structural change in the functional groups of alkali-induced FSG samples with or without curcumin loading. The FTIR spectra of native FSG and AFSG displayed broad characteristic absorption bands at 3429 cm^−1^ (FSG), 3438 cm^−1^ (AFSG_2h_), and 3442 cm^−1^ (AFSG_8h_), respectively (Figure 7A), indicating N-H stretching vibrations associated with hydrogen bonding within the wavenumber range of 3500 to 3000 cm^−1^ [31]. With the extension of alkali hydrolysis, the intensity of the peaks in AFSG showed a gradual reduction relative to native FSG, indicating that alkali hydrolysis likely impaired certain interchain hydrogen bonds, thereby altering the structural integrity of the protein. Meanwhile, a simultaneous shift in the amide A band of both FSG and AFSG samples was observed following curcumin loading, suggesting the establishment of hydrogen bonds between the ester and hydroxyl groups present in curcumin [8,37]. Another absorption band was detected at 2929 cm^−1^, attributed to the C-H stretching of methyl groups [18]. Notably, the prominent peak within the 2800–3000 cm^−1^ ranges in the FTIR spectra initially intensified and then weakened following the alkali hydrolysis of FSG, indicating that the hydrophobic interaction after alkali hydrolysis experienced an enhanced and then attenuated tendency. These results are aligned with the result of the intermolecular interaction (Figure 3). After the loading of curcumin, the CL-AFSG samples’ peak intensity was considerably higher compared to CL-FSG, likely due to the hydrophobic CH_2_ chains participating in non-covalent interactions with curcumin. Additionally, major characteristic peaks of FSG were observed at approximately 1649, 1546, and 1245 cm^−1^, corresponding to the amide I band (1700–1600 cm^−1^), amide II band (1600–1500 cm^−1^), and amide III band (1330–1220 cm^−1^), respectively [38]. The peak at 1649 cm^−1^ of AFSG is more intense than that for the FSG group and is related to amide I (1600–1700 cm^−1^), resulting from the stretching vibration of the C=O and C-N groups. Furthermore, the wave number corresponding to an α-helix is 1638–1664 cm^−1^ [14], and the characteristic absorption peak of α-helix shifted to a lower wavenumber from 1649 (FSG) to 1648 (AFSG_2h_) and 1636 (AFSG_8h_) cm^−1^, respectively. Hence, the role of α-helix in FSG weakened gradually with the prolongation of alkali hydrolysis; this result aligned with the findings from the CD spectrum. The peak at 1546 cm^−1^ in FSG corresponded to the amide II band, which represents C-N stretching combined with in-plane N-H bending modes and C-C stretching vibrations [39]. During alkali hydrolysis, the peak corresponding to the amide II band in AFSG gradually diminished and eventually disappeared. The altered intensity of the amide I and II bands could be attributed to the hydrophobic interactions involving C=O, C-N, and N-H groups during alkali hydrolysis. The peak observed at 1245 cm^−1^ in AFSG, exhibiting greater intensity compared to the FSG group, was linked to the amide III band. This band mainly arises from C-N stretching vibrations, N-H deformation of the amide bond, and the swinging vibrations of CH_2_ groups in glycine and proline residues [40]. Interestingly, the FTIR spectrum of CL-FSG closely matched that of FSG, displaying characteristic peaks at approximately 1718, 1599, 1401, and 1234 cm^−1^ (Figure 7B). This result strongly suggests that the primary structure of FSG remained essentially unchanged after the encapsulation of curcumin. In addition, as presented in Figure 7B–D, when curcumin was incorporated into FSG and AFSG samples, its characteristic peaks (722, 812, 961, 1151, 1276, 1507 cm^−1^) were clearly detected in the spectra. However, following curcumin loading, these distinct peaks nearly vanished, providing strong evidence of the efficient encapsulation of curcumin within the samples.

#### 3.8.2. XRD Analysis

To further examine the physical state of the samples, X-ray diffraction (XRD) analysis was conducted to reflect the change of these samples in crystalline structure or amorphous structure. As shown in Figure 8A, FSG presented one flat hump at diffraction angles 2θ of 19.40°, indicating the amorphous nature of the FSG. At the same time, the alkali-treated FSG showed a similar amorphous structure. In contrast, pure curcumin demonstrated a highly crystalline structure, with distinct and sharp diffraction peaks observed in the 2θ range of 6–30°, including 8.89°, 14.48°, 17.22°, 18.18°, 23.3°, 24.60°, and 25.52°. Additionally, as shown in Figure 8B–D, weak characteristic peaks including (8.89° and 17.22°) of curcumin were found in the pattern of physical mixture including FSG-curcumin (mix), AFSG2h-curcumin (mix) and AFSG8h-curcumin (mix), indicating that the crystalline state of curcumin is preserved in the combination of curcumin with FSG and AFSG. This occurs because the physical mixing process lacks sufficient molecular interactions between curcumin and FSG or AFSG, enabling curcumin to preserve its original crystalline structure [41]. However, after the loading of curcumin onto the gelatin matrix, the absence of curcumin’s crystalline peaks suggests its molecular dispersion within the gelatin matrix, facilitated by hydrophobic interactions and hydrogen bonding [10]. This results in the formation of a supramolecular complex through molecular interactions between curcumin and FSG or AFSG molecules rather than mere physical encapsulation, as evidenced by FTIR (Figure 7). Moreover, this transformation may also be attributed to intermolecular interactions that facilitated the formation of an amorphous state, where the resulting amorphization stems from successful curcumin encapsulation within the gelatin matrix [42]. Consistently, it has been reported that the encapsulation of curcumin within the zein matrix induces its transformation into an amorphous state [43].

#### 3.8.3. Fluorescence Spectrum Analysis

Fluorescence quenching serves as a useful technique for characterizing the interaction between proteins and small bioactive molecules. The analysis primarily targets the tyrosine residues and the changes in their surrounding environment at an excitation wavelength of 280 nm when small molecules interact with the gelatin [44]. Since protein and bioactive small molecules can cause fluorescence quenching and even the maximum emission wavelength shift, endogenous fluorescence quenching of proteins can be used to study the interactions between different treated FSG and hydrophobic bioactive compounds. Figure 9 displays the fluorescence spectra of FSG and AFSG across a range of curcumin concentrations (from 0 to 6 μmol/mL).

As curcumin concentration increased, a more significant reduction in fluorescence intensity was observed for FSG, AFSG_2h_, and AFSG_8h_, indicating that curcumin induced a quenching effect on the protein’s endogenous fluorescence. In addition, compared with FSG, alkali-treated FSG exhibited stronger fluorescence intensity without the addition of curcumin. This is attributed to alkali hydrolysis, which disrupts the native FSG molecular structure, leading to the exposure of additional interior hydrophobic groups, thereby increasing fluorescence intensity. It is worth noting that with the increased curcumin concentration, an evident blue shift occurred in all samples, which indicates a reduction in the microenvironment polarity surrounding the hydrophobic amino acids, suggesting that curcumin interacts with FSG or AFSG within a hydrophobic environment. With the enhancement of hydrophobic interactions, curcumin becomes incorporated into the hydrophobic pockets of FSG or AFSG.

Fluorescence quenching occurs when a fluorophore interacts with quencher molecules, resulting in a reduction of fluorescence intensity [26]. This phenomenon can be categorized into two types: static and dynamic quenching. Static quenching occurs when a stable ground-state complex forms between the fluorophore and quencher through coordination interactions. In contrast, dynamic quenching arises from molecular collisions or interactions between the fluorophore and quencher in the excited state, leading to a non-radiative energy transfer that diminishes fluorescence. To identify the predominant quenching mechanism (dynamic or static), a linear plot of the ratio F_0_/F versus quencher concentration, [Q], was plotted for analysis.

The calculated k_q_ values for FSG, AFSG_2h_, and AFSG_8h_ were 6.49 × 10^13^, 8.14 × 10^13^, and 6.65 × 10^13^ L/mol^−1^·s^−1^, respectively, which is much higher than the maximum dynamic quenching constant 2.0 × 10^10^ L/mol^−1^·s^−1^ for quenchers interacting with biopolymers [45]. It can be deduced that fluorescence quenching originated from a static mechanism caused by the gelatin-small molecule interaction; thus, non-luminescent complexes were formed. Concerning the number of binding sites and the binding constant (Figure 9), the double logarithmic curves of FSG and AFSG samples exhibit a strong linear correlation. According to Table 2, the K_a_ values were ordered as follows: AFSG_2h_ (1.07 × 10^7^ L·mol^−1^), AFSG_8h_ (9.19 × 10^5^ L·mol^−1^), and FSG (8.43 × 10^4^ L·mol^−1^). The higher K_a_ values reflected a stronger binding between the protein and curcumin molecules [46]. Notably, AFSG_2h_-curcumin demonstrated the most robust binding, while FSG-curcumin exhibited the weakest interaction. These findings suggest that alkali hydrolysis, particularly at the optimal level, facilitated the self-assembly of FSG by promoting the aggregation of hydrophobic groups, which in turn enhanced the hydrophobic interactions. Additionally, the binding site numbers for FSG with curcumin were approximately 0.83, while AFSG_2h_ and AFSG_8h_ exhibited values of 1.21 and 1.02, respectively. This suggests that FSG, after alkali hydrolysis, binds to a single class of sites on curcumin, with AFSG_2h_ showing the highest number of binding sites. These results align with the solubilization behavior of curcumin in FSG and AFSG samples.

#### 3.8.4. CLSM

To gain deeper insight into the interaction mechanism between curcumin and FSG or AFSG samples, CLSM was employed to provide a detailed characterization of the morphology of the prepared CL-FSG and CL-AFSG nanoparticles. Figure 10 revealed that the FSG possessed a larger size than those in AFSGs via individual channel images with heterogenous distribution, considering the FSG was decomposed into pieces (with smaller M_w_) after the alkali hydrolysis. However, interestingly, after the alkali hydrolysis for 2 h, the nanoparticles displayed a fairly even distribution, with curcumin nanoparticles colocalizing within the FSG or AFSG_2h_ matrix (labeled with rhodamine B), demonstrating uniform loading of curcumin nanoparticles in the AFSG_2h_ matrix. However, when the alkali hydrolysis was elongated up to 8 h, the AFSG_8h_ matrix and curcumin nanoparticles were not fully merged, with a small quantity of curcumin scattered around the solvent, suggesting ineffective loading of curcumin nanoparticles within the AFSG_8h_ nanoparticle.

## 4. Conclusions

This study elucidated the structural and functional modifications of fish scale gelatin induced by alkaline hydrolysis. Alkali-induced degradation of gelatin yields different polymers with molecular weights (M_w_) from 19,319 to 3881 Da. The optimal loading efficiency of curcumin was observed at alkali-induced hydrolysis of FSG for 2 h. TEM and DLS showed a particle size of 1266.3 nm and PDI of 0.824 in terms of FSG in an aqueous system, whereas alkali hydrolysis significantly reduced both the particle size and PDI of AFSG. The incorporation of curcumin into the AFSG assembly strengthened hydrophobic interactions, leading to a notable reduction in particle size and PDI and a decrease in ζ-potential due to the adsorption of adjacent AFSGs onto the surface. Moderate hydrolysis treatment facilitated the release of hydrophobic amino acids within AFSG_2h_, promoting the exposure of hydrophobic regions and the formation of soluble clusters. The strengthened hydrophobic interactions and higher β-sheet content collectively increased the structural flexibility, enabling AFSG_2h_ to expose additional hydrophobic groups. CLSM and fluorescence spectroscopy revealed that AFSG_2h_ was easier to interact with curcumin, which improved the corresponding loading efficiency of curcumin. In contrast, excessive hydrolysis of fish scale gelatin led to significant structural degradation and severe aggregation, which increased particle size and reduced PDI. Moreover, it weakened hydrophobic interactions, which in turn lowered the curcumin loading efficiency. In summary, optimal alkali hydrolysis effectively enhances the curcumin loading efficiency of fish scale gelatin, offering a green and facile approach for developing protein-based nanoparticles. These findings optimize the functionality of alkali-treated fish scale gelatin as prospective nanocarriers, extending its potential uses.

## Figures and Tables

**Figure 1 foods-14-01183-f001:**
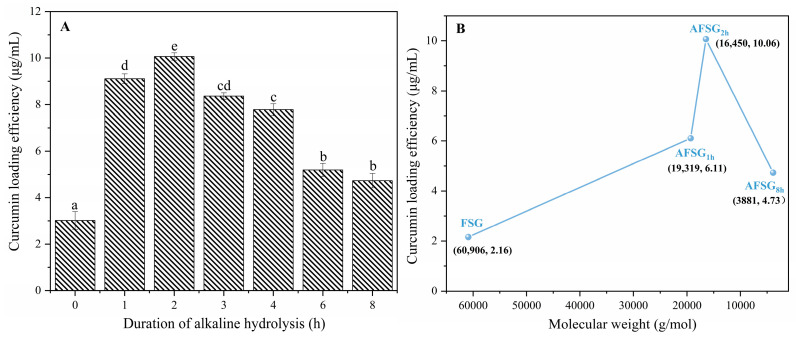
Curcumin loading efficiency of fish scale gelatin (FSG) with 0–8 h alkali-induced hydrolysis (**A**) and alkali-induced fish scale gelatin (AFSG) with different molecular weights and duration of alkali hydrolysis (**B**). Different lowercase letters indicate significant differences in the alkali hydrolysis group, respectively (*p* < 0.05).

**Figure 2 foods-14-01183-f002:**
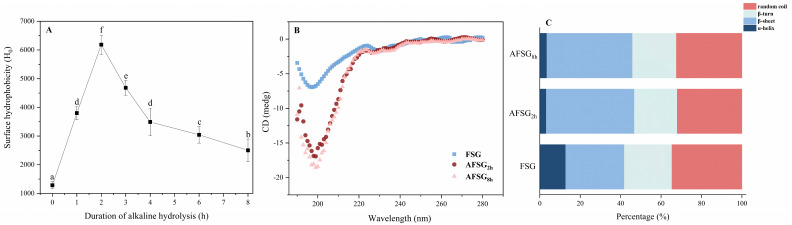
Surface hydrophobicity (**A**), different superscript letters indicate significant differences at the *p* < 0.05 level at each alkaline hydrolysis level, circular dichroism spectra (**B**), and secondary structure composition (**C**) of FSG and AFSGs.

**Figure 3 foods-14-01183-f003:**
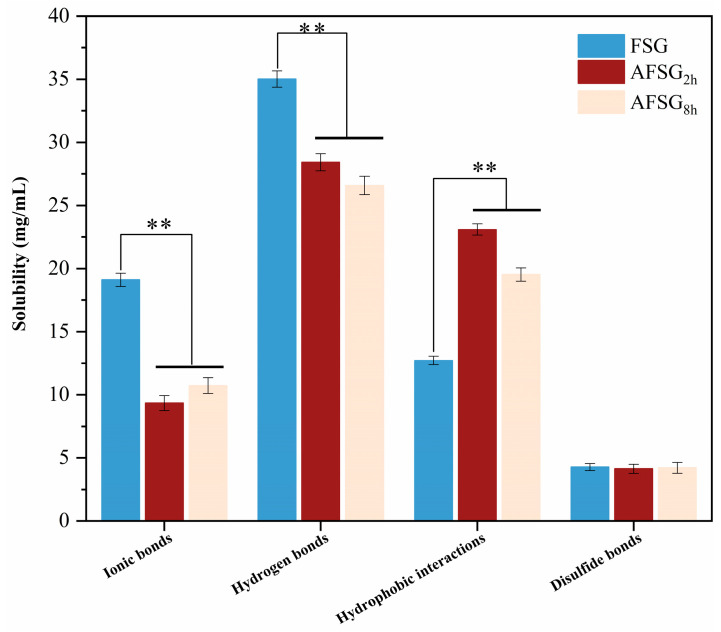
Intermolecular interaction of FSG and AFSGs (AFSG_2h_, duration of alkali hydrolysis = 2 h; AFSG_8h_, duration of alkali hydrolysis = 8 h). Data bearing double asterisks are significantly different (*p* < 0.01) between AFSG and FSG in the same chemical interaction.

**Figure 4 foods-14-01183-f004:**
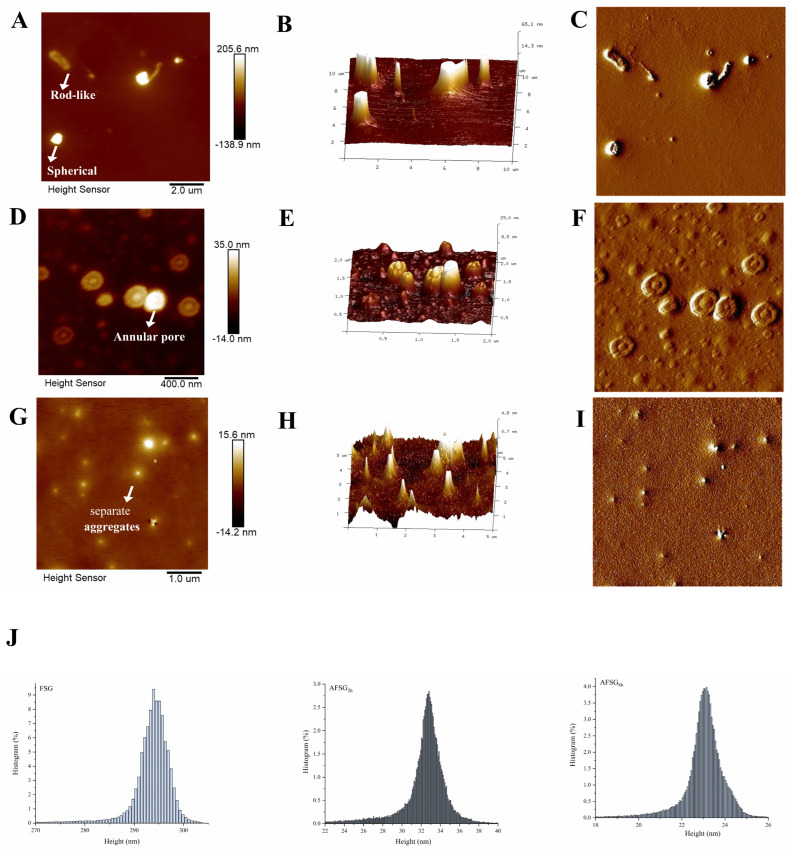
Atomic force microscopy of fish scale gelatin (FSG) and AFSGs. (**A**–**C**) Two−dimensional height mode image, three-dimensional version of the corresponding two-dimensional height mode image and error signal mode image of FSG; (**D**–**F**) two-dimensional height mode image, three-dimensional version of the corresponding two-dimensional height mode image and error signal mode image of AFSG_2h_; (**G**–**I**) two-dimensional height mode image, three-dimensional version of the corresponding two-dimensional height mode image and error signal mode image of AFSG_8h_. (**J**) Effects of duration of alkaline hydrolysis on the distribution of height of FSG and AFSG.

**Figure 5 foods-14-01183-f005:**
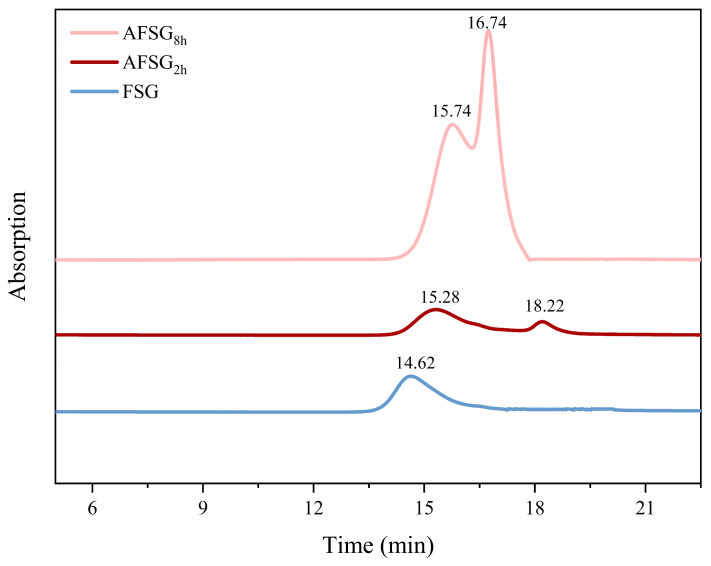
Molecular weight distribution of FSG and AFSGs (AFSG_2h_, duration of alkali hydrolysis = 2 h; AFSG_8h_, duration of alkali hydrolysis = 8 h).

**Figure 6 foods-14-01183-f006:**
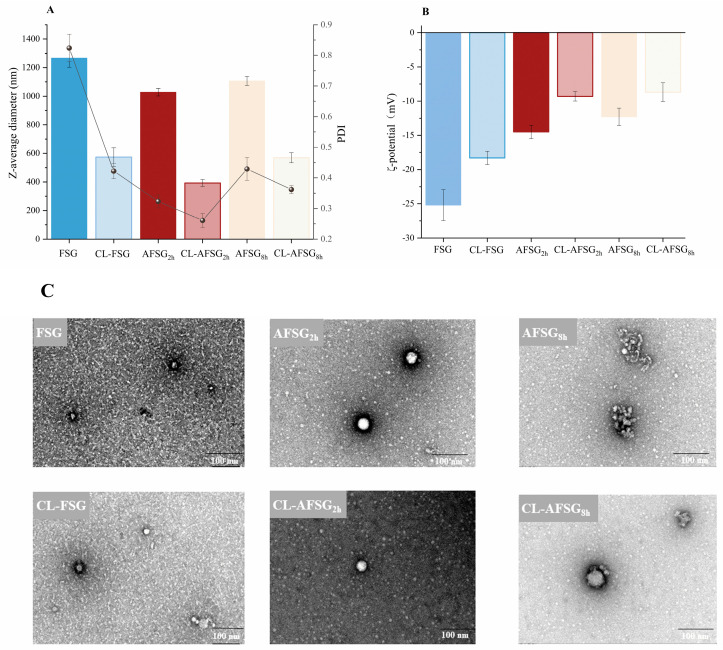
Particle size and particle distribution index (PDI) (**A**) and ζ−potential (**B**,**C**) of FSG, AFSGs, and curcumin-loaded FSG and AFSG (CL-FSG, CL-AFSG).

**Figure 7 foods-14-01183-f007:**
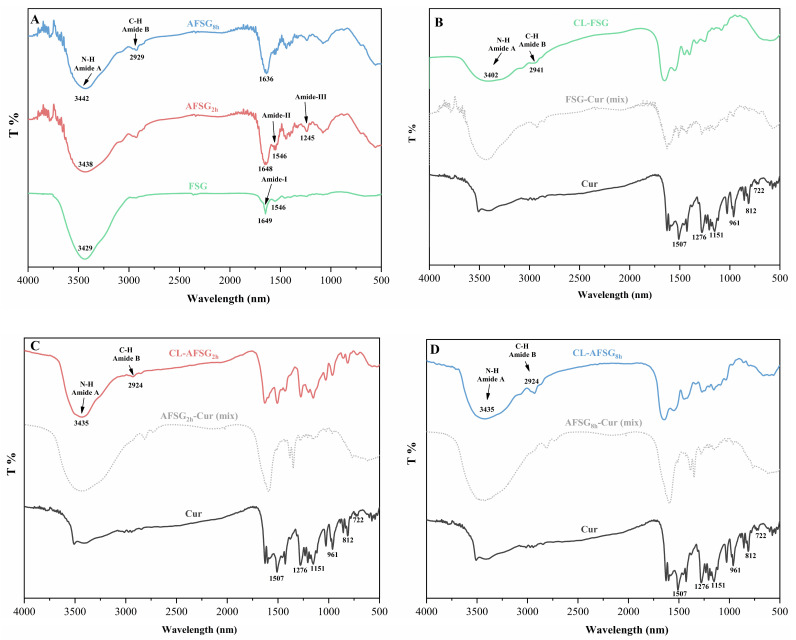
FT-IR spectra of fish scale gelatin (FSG) and alkali-induced fish scale gelatin (AFSG) with or without loading curcumin. (**A**) Comparison of FT-IR spectra for FSG vs AFSG at 2 h and 8 h of alkali hydrolysis. (**B**–**D**) FT-IR spectra of: (**B**) native FSG, (**C**) 2 h alkali-hydrolyzed AFSG, and (**D**) 8 h alkali-hydrolyzed AFSG, showing: (i) unloaded samples, (ii) curcumin-loaded samples, (iii) physical mixtures of FSG/AFSG and curcumin, and (iv) pure curcumin control.

**Figure 8 foods-14-01183-f008:**
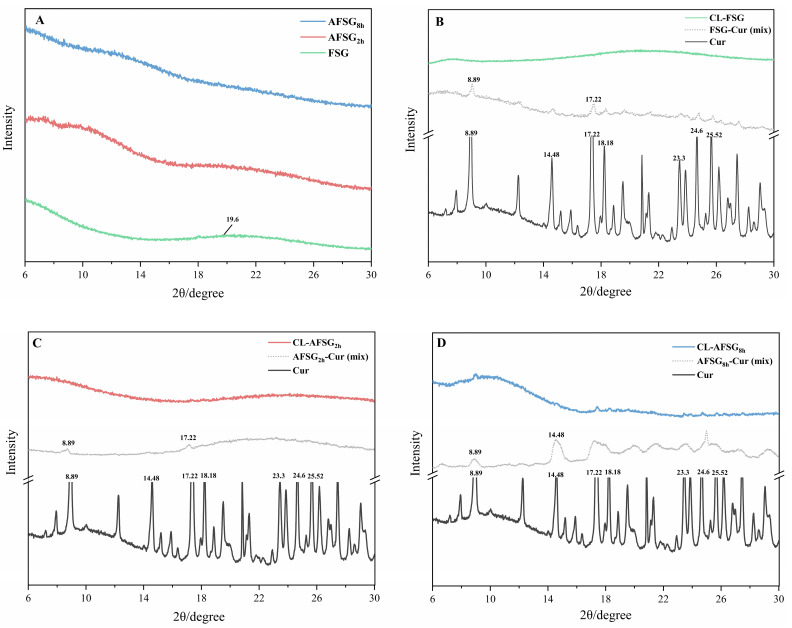
X-ray diffraction (XRD) of fish scale gelatin (FSG) and alkali-induced fish scale gelatin (AFSG) with or without loading curcumin and their mixture. (**A**) Comparison of XRD for FSG vs AFSG at 2 h and 8 h of alkali hydrolysis. (**B**–**D**) XRD of: (**B**) native FSG, (**C**) 2 h alkali-hydrolyzed AFSG, and (**D**) 8 h alkali-hydrolyzed AFSG, showing: (i) unloaded samples, (ii) curcumin-loaded samples, (iii) physical mixtures of FSG/AFSG and curcumin, and (iv) pure curcumin control.

**Figure 9 foods-14-01183-f009:**
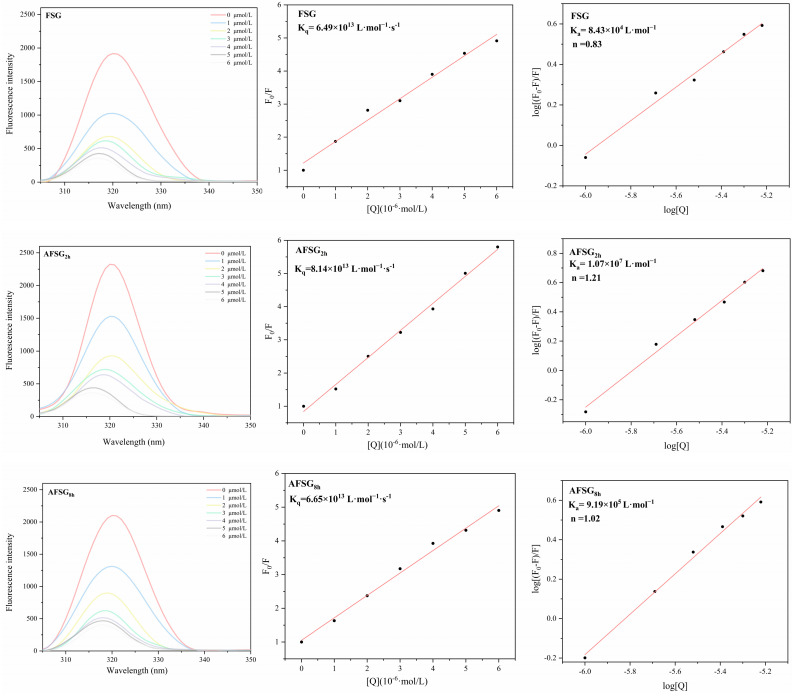
The fluorescence spectra of FSG and AFSG (alkali-induced FSG) with different concentrations of curcumin (0−6 μmol/L) and corresponding Stern–Volmer and double-logarithmic regression plots for the quenching of FSG and AFSG by curcumin.

**Figure 10 foods-14-01183-f010:**
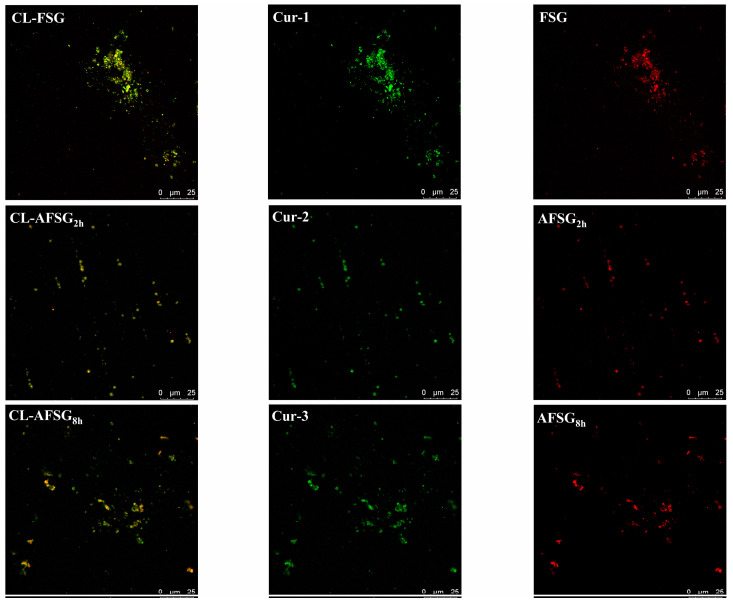
CLSM images of FSG (fish scale gelatin) and AFSGs (with alkali hydrolysis for 2 h, AFSG_2h_ and alkali hydrolysis for 8 h, AFSG_8h_).

**Table 1 foods-14-01183-t001:** Amino acid content of FSG, AFSG_2h_, and AFSG_8h_.

Amino Acids	Content/(mg/g)		
	FSG	AFSG_2h_	AFSG_8h_
Aspartic acid (Asp)	45.55 ± 0.13 ^b^	37.24 ± 0.05 ^a^	35.44 ± 0.07 ^a^
Threonine (Thr)	21.91 ± 0.07 ^b^	18.38 ± 0.04 ^a^	17.06 ± 0.06 ^a^
Serine (Ser)	28.22 ± 0.05 ^b^	23.49 ± 0.12 ^a^	26.19 ± 0.24 ^a b^
Glutamic acid (Glu)	61.06 ± 0.24 ^c^	49.72 ± 0.23 ^a^	53.62 ± 0.46 ^b^
Glycine (Gly)	84.91 ± 0.32 ^b^	74.29 ± 0.17 ^a^	72.54 ± 0.22 ^a^
Alanine (Ala)	54.97 ± 0.09 ^b^	56.83 ± 0.07 ^a^	55.93 ± 0.19 ^a^
Cystine (Cys)	2.16 ± 0.01 ^b^	2.62 ± 0.12 ^b^	1.59 ± 0.45 ^a^
Valine (Val)	15.62 ± 0.06 ^b^	16.63 ± 0.25 ^a^	16.25 ± 0.13 ^a^
Methionine (Met)	20.16 ± 0.05 ^a^	21.19 ± 0.12 ^a^	20.38 ± 0.37 ^a^
Isoleucine (Ile)	24.64 ± 0.03 ^a^	26.39 ± 0.13 ^a^	25.85 ± 0.27 ^a^
leucine (Leu)	33.75 ± 0.16 ^a^	33.98 ± 0.24 ^a^	32.64 ± 0.13 ^a^
Tyrosine (Tyr)	6.54 ± 0.07 ^a^	6.31 ± 0.14 ^a^	6.09 ± 0.21 ^a^
Phenylalanine (Phe)	15.44 ± 0.03 ^a^	16.98 ± 0.26 ^a^	17.26 ± 0.53 ^a^
Histidine (His)	7.99 ± 0.09 ^a^	7.43 ± 0.14 ^a^	7.13 ± 0.23 ^a^
Lysine (Lys)	28.89 ± 0.14 ^c^	26.83 ± 0.25 ^b^	23.53 ± 0.39 ^a^
Arginine (Arg)	43.26 ± 0.23 ^c^	36.68 ± 0.15 ^a^	39.93 ± 0.36 ^b^
Proline (Pro)	29.15 ± 0.74 ^a^	34.02 ± 0.67 ^c^	32.12 ± 0.38 ^b^
total hydrophobic amino acids (HAA) *	215.63 ± 2.16 ^c^	206.03 ± 2.33 ^b^	200.43 ± 2.12 ^a^
total amino acids (AA)	524.21 ± 2.39 ^c^	489.02 ± 1.83 ^b^	483.55 ± 2.46 ^a^
ratio of HAA to AA%	41.13 ± 1.84 ^a^	42.13 ± 2.43 ^b^	41.45 ± 1.95 ^a^

* Ala, Val, Leu, Ile, Phe, Pro, Thr and Met. Values represent the mean (n = 3) ± SD. Different letters in the same row represent significant differences among samples (*p* < 0.05).

**Table 2 foods-14-01183-t002:** Stern-Volmer quenching constants, the binding constant, and the numbers of binding sites of fish scale gelatin (FSG) and alkali-induced fish scale gelatin (AFSG) ^1,2^.

Sample	K_SV_ × 10^5^ (L·mol^−1^)	k_q_ × 10^13^ (L·mol^−1^·s^−1^)	K_a_ (L·mol^−1^)	n
FSG	6.49	6.49	8.43 × 10^4^	0.83
AFSG_2h_	8.14	8.14	1.07 × 10^7^	1.21
AFSG_8h_	6.65	6.65	9.19 × 10^5^	1.02

^1^ K_SV_ and k_q_ are the Stern-Volmer constant and the biomolecular quenching constant, which can be obtained by linear regression of a plot of F_0_/F against [Q]. ^2^ K_a_ describes the binding constant of FSG and AFSG with curcumin, while n is the number of binding sites.

## Data Availability

The original contributions presented in this study are included in the article/Appendix A. Further inquiries can be directed to the corresponding author.

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
