# Peer review of "Alkali-Induced Hydrolysis Facilitates the Encapsulation of Curcumin by Fish (Cyprinus carpio L.) Scale Gelatin"

_foods, 2025, doi:10.3390/foods14071183_

Round 1

Reviewer 1 Report

Comments and Suggestions for Authors

Dear Authors,
Thanks for studying the loading of curcumin to alkali induced fish scale gelatin. The manuscript is interesting, and it is well organized and written. However, I have these suggestions that may enhance article's potential readability:
1) A wider description of the importance of loading curcumin into alkali induced fish scale gelatin is suggested, to strengthen the relevance of the findings shown in this manuscript. Readers may think that there are currently plenty of dishes using curcumin as an ingredient, without degradation neither during the processing nor during the storage in dry conditions.
2) It is recommended to avoid undefined acronyms in the abstract (i.e., AFSG2h or FSG).
3) It is suggested standardizing the decimal significative numbers, according to their corresponding errors, in Table 1. Probably two decimals (mg/g) in every measurement would be fine, both in means and errors.
4) An explanation about why different AFM image lateral range values were selected is recommended (i.e., scale bars of A, D, and G images are different).
5) Better background subtraction in AFM pictures is suggested. It seems that line-to-line linear subtraction was made, leaving black non-real lines where a higher peak is found. Only the baseline should be considered in the subtraction process. This also affects the histograms.
Best regards.

Comments on the Quality of English Language

Typos found:
* "However," in line 62.
* "(" missing in line 125.
* "H0" in lines 160 and 330.
* "−1" in lines 201, 616, and 617.
* "The" in line 342.
* "Ka" in line 616.
* Extra " in line 627.
* "AFSG8h" in line 644.

Author Response

Reviewer #1: General comments

Thanks for studying the loading of curcumin to alkali induced fish scale gelatin. The manuscript is interesting, and it is well organized and written. However, I have these suggestions that may enhance article's potential readability.

Overall response: We truly appreciate your observations and value your supportive reaction to the nature of our work. We are grateful for the time and energy you put into developing this review. We are hopeful that you will agree that the revision is much improved and the theoretical and empirical contributions are much stronger.

Specific comments

  1. A wider description of the importance of loading curcumin into alkali induced fish scale gelatin is suggested, to strengthen the relevance of the findings shown in this manuscript. Readers may think that there are currently plenty of dishes using curcumin as an ingredient, without degradation neither during the processing nor during the storage in dry conditions.

Response: Thank you for your valuable suggestion. In the revised manuscript, we have expanded the discussion on the importance of loading curcumin into alkali-induced fish scale gelatin to highlight the relevance of our findings further. The related sentence was supplemented as “While curcumin is widely used in culinary applications, its inherent limitations—such as rapid photodegradation, poor aqueous solubility (∼11 ng/mL in water), and thermal instability during cooking processes—severely restrict its bioavailability and industrial applicability [1]. Thus, we hypothesized that the encapsulation within alkali-induced fish scale gelatin (AFSG) offers a promising strategy to enhance curcumin’s dispersibility in aqueous environments while providing structural protection against degradation. By investigating the encapsulation of curcumin within AFSG, this study provides insights into an alternative delivery system that extends curcumin’s applications beyond conventional culinary use.” Please see lines 83 – 91. Again, we appreciate your insightful comments, which have helped strengthen the clarity and significance of our research.

  1. It is recommended to avoid undefined acronyms in the abstract (i.e., AFSG2h or FSG).

Response: Thank you for your suggestion. We have revised the abstract to remove acronyms such as AFSG2h and FSG to ensure clarity and avoid ambiguity for readers.

  1. It is suggested standardizing the decimal significative numbers, according to their corresponding errors, in Table 1. Probably two decimals (mg/g) in every measurement would be fine, both in means and errors.

Response: Thank you for your suggestion. We have standardized the decimal significant figures in Table 1 by reporting all measurements and corresponding errors with two decimal places (mg/g), ensuring consistency throughout the table.

  1. An explanation about why different AFM image lateral range values were selected is recommended (i.e., scale bars of A, D, and G images are different).

Response: Thank you for your suggestion. Uniform scaling was avoided to prevent loss of detail resolution in structurally heterogeneous regions, as standardized ranges would compromise either fine feature clarity (overscaling) or representative field-of-view (underscaling). Specifically, the scale bars for images A, D, and G were adjusted to capture distinct morphological features optimally at appropriate resolutions. For instance, a larger lateral range in image A was used to depict the overall topography, while the smaller ranges in images D and G were chosen to highlight finer surface details. This multi-scale imaging strategy follows established protocols for nanomaterial characterization [2-3]. Therefore, to clarify the reason for selecting different AFM image lateral range values, we have added an explanation in the revised manuscript in Section 2.8. The related sentence has been supplemented as follows: “To optimally capture distinct morphological features of the samples, the scanning rate in auto-mode was dynamically adjusted based on scan size, ensuring a constant pixel resolution of 512 × 512 across all images [3].” Please see lines 208 – 211.

  1. Better background subtraction in AFM pictures is suggested. It seems that line-to-line linear subtraction was made, leaving black non-real lines where a higher peak is found. Only the baseline should be considered in the subtraction process. This also affects the histograms.

Response: We appreciate your insightful observation regarding background subtraction in the AFM images. We have re-evaluated our AFM image processing and revised the background subtraction procedure. The revised Figure 4 eliminates the artificial "black" lines previously observed in areas with higher peaks. Additionally, the histograms have been updated to represent the true surface morphology.

  1. Comments on the Quality of English Language Typos found: "However," in line 62. * "(" missing in line 125. * "H0" in lines 160 and 330. * "−1" in lines 201, 616, and 617. * "The" in line 342. * "Ka" in line 616. * Extra " in line 627. * "AFSG8h" in line 644.

Response: Thank you for your meticulous review of the manuscript’s language quality. We have corrected the noted issues accordingly. Furthermore, the full text has been rechecked and manually proofread to eliminate residual inconsistencies. 

Reference Used in Responses to Reviewer 1

  1. Lin, D.; Xiao, L.; Qin, W.; Loy, D. A.; Wu, Z.; Chen, H.; Zhang, Q. Preparation, characterization and antioxidant properties of curcumin encapsulated chitosan/lignosulfonate micelles. Carbohydrate Polymers 2022, 281, 119080.
  2. Dai, L.; Sun, C.; Li, R.; Mao, L.; Liu, F.; Gao, Y. Structural characterization, formation mechanism and stability of curcumin in zein-lecithin composite nanoparticles fabricated by antisolvent co-precipitation. Food Chemistry 2017, 237, 1163-1171.
  3. Chen, F.-P.; Li, B.-S.; Tang, C.-H. Nanocomplexation of soy protein isolate with curcumin: Influence of ultrasonic treatment. Food Research International 2015, 75, 157-165.

Reviewer 2 Report

Comments and Suggestions for Authors

The document presents a novel approach to obtaining nanoparticles that act as reservoirs for curcumin through encapsulation and intermolecular interactions. However, the document requires some improvements, which are outlined below:

Introduction Revisions: The introduction contains several issues that need correction.

Pathogen Risk Clarification: The authors mention that mammalian gelatin poses a risk of pathogens, but it should be noted that fish gelatin also presents this issue, particularly in freshwater fish.

Drafting Improvements: Minor drafting improvements are needed throughout the document.

Section 2.11 Clarification: Section 2.11 is unclear and requires revision.

Figure Quality: The figures have poor quality and should be improved.

Abbreviation Consistency: Use different abbreviations for solutions (e.g., S1...) to avoid confusion with supplementary materials (e.g., Table S1).

Materials and Reagents List: The list of materials and reagents is incomplete and needs to be corrected.

Crystalline Structure Formation: Clarify how the crystalline structure of curcumin was formed.

Encapsulation Process Clarification: In paragraph line 526, clarify whether you are referring to an encapsulation process or the formation of a supramolecular complex

Author Response

Reviewer #2: General comments

The document presents a novel approach to obtaining nanoparticles that act as reservoirs for curcumin through encapsulation and intermolecular interactions. Drafting Improvements: Minor drafting improvements are needed throughout the document. However, the document requires some improvements, which are outlined below:

Overall response: We appreciate your recognition of our study's novelty in developing nanoparticles as curcumin reservoirs. We have carefully reviewed the manuscript and implemented the necessary drafting improvements to enhance clarity and readability. Additionally, we have addressed the specific revisions outlined in your comments to further refine the document.

Reviewer #2: Specific comments

1.Introduction Revisions: The introduction contains several issues that need  correction. Pathogen Risk Clarification: The authors mention that mammalian gelatin poses a risk of pathogens, but it should be noted that fish gelatin also presents this issue, particularly in freshwater fish.

Response: Thank you for your insightful comment. We sincerely apologize for the lack of clarity in our discussion of pathogen risks. Our intention was to highlight the specific concern of prion-related pathogens in mammalian gelatin rather than suggesting that all pathogens are exclusive to mammalian sources. We acknowledge that fish gelatin, particularly from freshwater species, may also present microbial contamination risks. To address this, we have revised the introduction to better distinguish between the risks associated with mammalian and fish gelatin, ensuring a more precise and balanced discussion. The related sentence has been supplemented as “Noticeably, mammalian sources account for about 98.5% of the world’s gelatin production [13]. However, the safety of mammalian gelatin has been increasingly questioned due to its potential role in transmitting prion-related pathogens, particularly in the wake of diseases such as bovine spongiform encephalopathy (BSE) and foot-and-mouth disease (FMD) [14]. In response to these concerns, global consumers with strict adherence to vegetarianism and religious beliefs seek alternative protein sources with similar functionalities for food systems [12]. As a result, there has been growing interest in alternative gelatin sources, particularly those derived from freshwater fish processing by-products like scales, bones, and skin. Compared to mammalian gelatin, these alternatives not only eliminate concerns associated with prion-related diseases but also better align with the dietary preferences and ethical considerations of consumers adhering to vegetarianism and religious restrictions.
Although freshwater fish may be susceptible to certain microbial contaminants, appropriate processing and purification methods can effectively mitigate these risks, ensuring its safety and suitability as a viable alternative to mammalian gelatin in food applications [12].” Please see Lines 55 – 69.

2.Section 2.11 Clarification: Section 2.11 is unclear and requires revision.

Response: Thank you for your valuable feedback. In the revised manuscript, we have clarified Section 2.11 by providing a more detailed description of the methodology, ensuring clearer explanations of the procedures and parameters used. The related sentence was supplemented as “The spatial distribution of curcumin-loaded nanoparticles in native and alkali-modified FSG matrices was analyzed via CLSM (LEICA TCS SP8) following established protocols [1]. Fluorescence detection parameters were configured as follows: Curcumin fluorescence was detected at 470–556 nm (green channel) under 405 nm excitation; rhodamine B-labeled gelatin samples (both native and alkali-treated FSG) were analyzed using 570 nm excitation with emission collected at 565–685 nm (red channel). Following standardized preparation, samples were deposited on 1 mm glass substrates and equilibrated at 10 °C for 12 h prior to imaging. Image acquisition was conducted using a HyD high-sensitivity detector with argon-ion and He-Ne laser.” Please see Lines 269 – 277.

3.Figure Quality: The figures have poor quality and should be improved.

Response: Thank you for pointing out the quality issues with the uploaded Figures. We carefully check the quality of all figures throughout the manuscript to ensure they meet the required standards for clarity and readability. We re-uploaded the figures with high definition, this should now be solved.

  1. Abbreviation Consistency: Use different abbreviations for solutions (e.g., S1...) to avoid confusion with supplementary materials (e.g., Table S1).

Response: Thank you for highlighting this critical issue of abbreviation consistency. We have revised all instances where solution numbering (e.g., S1, S2) conflicted with supplementary material references. Solutions are now labeled as Sol1, Sol2, etc (bolded in Methods Section 2.7), while supplementary tables/figures retain Table S1, Figure S1 notation. No overlapping abbreviations remain in the main text or supplements.

  1. Materials and Reagents List: The list of materials and reagents is incomplete and needs to be corrected.

Response: Thank you for your valuable suggestion. We apologize for the oversight. In the revised manuscript, detailed information was added within the Material and Methods Section 2.1. The related sentence was supplemented as “Sodium chloride (NaCl), sodium hydroxide (NaOH), hydrochloric acid (HCl), potassium bromide (KBr) of infrared (IR) spectroscopy grade, and β-mercaptoethanol (β-ME) were obtained from Macklin Biochemical Co., Ltd. (Shanghai, China). Coomassie Brilliant Blue was bought from Servicebio Biotechnology Co., Ltd. (Wuhan, China). Curcumin (98%) was supplied by Adamas Reagent Co., Ltd. (China). 8-anilino-1-napthalenesulfonic acid ammonium salt (ANS) was acquired from Wan-bang Chemical Technology Co., Ltd (Nanjing, China).” Please see lines 113 – 119.

  1. Crystalline Structure Formation: Clarify how the crystalline structure of curcumin was formed. Encapsulation Process Clarification: In paragraph line 526, clarify whether you are referring to an encapsulation process or the formation of a supramolecular complex

Response: Thank you for your valuable comments. In the revised manuscript, we point out the reason for the crystalline structure of curcumin being retained in the physical mixture of FSG with or without alkali treatment and curcumin: “This occurs because the physical mixing process lacks sufficient molecular interactions between curcumin and FSG or AFSG, enabling curcumin to preserve its original crystalline structure [2] (Line 582 – 585).” Secondly, regarding the query on the encapsulation process or the formation of a supramolecular complex, our revised manuscript in Section 3.8.2 indicates that “However, after the loading of curcumin onto the gelatin matrix, the absence of curcumin’s crystalline peaks suggests its molecular dispersion within the gelatin matrix, facilitated by hydrophobic interactions and hydrogen bonding [10]. This results in the formation of a supramolecular complex through molecular interactions between curcumin and FSG or AFSG molecules rather than mere physical encapsulation, as evidenced by FTIR (Figure 7). Moreover, this transformation may also be attributed to intermolecular interactions that facilitated the formation of an amorphous state, where the resulting amorphization stems from successful curcumin encapsulation within the gelatin matrix [3] (Line 585 – 593).”

Reference Used in Responses to Reviewer 2

  1. Li, M.; Liu, Y.; Liu, Y.; Zhang, X.; Han, D.; Gong, J. pH-driven self-assembly of alcohol-free curcumin-loaded zein-propylene glycol alginate complex nanoparticles. International Journal of Biological Macromolecules 2022, 213, 1057-1067.
  2. Liu, J.; Li, H.; Wan Mustapha, W. A.; Zhang, X. Ultrasound-assisted alkaline hydrolysis of fish(Cyprinus carpio L.) scale gelatin: A possible curcumin delivery system with increased water solubility and sustained release. LWT-Food Science and Technology 2024, 191, 115589.
  3. Li, H.; Mustapha, W. A. W.; Liu, J.; Zhang, X. Self-assembled nanoparticles of acid-induced fish (Cyprinus carpio L.) scale gelatin: Structure, physicochemical properties, and application for loading curcumin. Food Chemistry: X 2024, 21, 101230.